# Physicochemical Properties, Fatty Acid Composition, Volatile Compounds of Blueberries, Cranberries, Raspberries, and Cuckooflower Seeds Obtained Using Sonication Method

**DOI:** 10.3390/molecules26247446

**Published:** 2021-12-08

**Authors:** Dorota Bederska-Łojewska, Marek Pieszka, Aleksandra Marzec, Magdalena Rudzińska, Anna Grygier, Aleksander Siger, Katarzyna Cieślik-Boczula, Sylwia Orczewska-Dudek, Władysław Migdał

**Affiliations:** 1Department of Animal Nutrition and Feed Sciences, National Research Institute of Animal Production, Krakowska Str. 1, 32-083 Balice, Poland; dorota.bederska@iz.edu.pl (D.B.-Ł.); sylwia.orczewska@iz.edu.pl (S.O.-D.); 2Liposolutions Ltd., Gospodaercza Street 26, 20-213 Lublin, Poland; office@liposulutions.pl; 3Faculty of Food Science and Nutrition, Poznan University of Life Sciences, Wojska Polskiego 28, 60-637 Poznan, Poland; magdalena.rudzinska@up.poznan.pl (M.R.); anna.grygier@up.poznan.pl (A.G.); asiger@up.poznan.pl (A.S.); 4Faculty of Chemistry, University of Wroclaw, ul. F. Joliot-Curie 14, 50-383 Wrocław, Poland; katarzyna.cieslik@chem.uni.wroc.pl; 5Department of Animal Product Technology, Balicka 122, 31-149 Kraków, Poland; wladyslaw.migdal@urk.edu.pl

**Keywords:** seed oils, fatty acids, tocopherols, tocotrienols, phytosterols

## Abstract

Every year, thousands of tons of fruit seeds are discarded as agro-industrial by-products around the world. Fruit seeds are an excellent source of oils, monounsaturated fatty acids, and n-6 and n-3 polyunsaturated essential fatty acids. This study aimed to develop a novel technology for extracting active substances from selected seeds that were obtained after pressing fruit juices. The proposed technology involved sonification with the use of ethyl alcohol at a low extraction temperature. Seeds of four species—blueberry (*Vaccinium myrtillus* L.), raspberry (*Rubus idaeus*), cranberry (*Vaccinium macrocarpon*), and cuckooflower (*Cardamine pratensis*)—were used for extraction. Following alcohol evaporation under nitrogen, the antioxidant activity, chemical composition, and volatile compounds of the obtained extracts were analyzed using chromatographic methods, including gas chromatography (GC)–mass spectrometry (MS) (GC–MS/MS), and high-performance liquid chromatography–MS. We analyzed physicochemical properties, fatty acid, and volatile compounds composition, sterol and tocochromanol content of blueberry, cranberry, raspberry, and cuckooflower seed oils obtained by sonication. This method is safe and effective, and allows for obtaining valuable oils from the seeds.

## 1. Introduction

Poland is a major fruit producer in the European Union. It ranks next to Spain, and outperforms France and Greece in fruit harvest. Fruit production is an important agricultural sector in the country. Fruit cultivation in Poland accounted for 13% of the overall crop production in 2019 (over 4 million PLN) and is on the rise, constituting approximately 10% of the total production volume of the European Union [1]. Seeds remaining after juice pressing are rich in valuable lipid compounds, and can therefore be considered as useful raw materials. Such seeds are characterized by high fat contents, and contain substances of high biological value. In addition, they are a valuable source of natural antioxidants (anthocyanins, polyphenols, flavonoids, vitamin E, beta-carotene, and xanthophylls), unsaturated fatty acids, mineral and aromatic compounds, antibacterial and antiviral substances, and so on [2]. As consumers prefer high-quality foods, there is a need to produce minimally processed food products, without the addition of chemical additives or synthetic vitamins. Supplementation with natural oils that are rich in the above-mentioned substances may improve the palatability and rheological properties of food products, as well as enhance their nutritional value. Among the lipid components in oils, triacylglycerols (TAGs; esters of glycerol and fatty acids) are the most important, followed by non-TAG groups of compounds such as phospholipids, sterols, tocopherols, and carotenoids, which are present in much smaller amounts [2,3]. These components determine the nutritional value of oils and influence their stability properties, especially oxidative stability. Seeds of berries, such as blueberries, cranberries, raspberries, and cuckooflower contain a high amount of PUFA. These acids are not synthesized in the human body, and hence must be supplied through food.

According to the recommendations of human nutrition specialists, the intake of total fats should be reduced. This can be achieved by increasing the intake of PUFA [4]. Apart from linoleic acid (LA) and long-chain polyunsaturated fatty acids (LC PUFA), α-linolenic acid (ALA), belonging to the family of n-3 acids, is an important PUFA. Its content in the studied oils is as high as 30%. A recent study reported that n-3 fatty acids have physiological and health-promoting properties, and are especially valuable in the prevention of cardiovascular diseases [5]. The health benefits of γ-linolenic acid (GLA) have also been reported, with a particular emphasis on the prevention of inflammatory and allergic diseases, and cardiovascular disorders [6].

Vegetable fatty acids, which belong to the group of essential unsaturated fatty acids (EUFA), are characterized by high biological activity. Seeds and oils derived from blueberries, cranberries, raspberries, and cuckooflowers contain the above-mentioned fatty acids, as well as antioxidant, anti-inflammatory, antiatherosclerotic, and anticancer substances, including tocochromanols, carotenoids, flavonoids, phytosterols, and phenolic acids [2,7]. Knowledge about the occurrence of antioxidants that can inhibit adverse changes in food and their activity and stability is important for technologists and nutritionists.

It is necessary to ensure that oils are properly protected after their extraction. Various measures taken for the protection of oils include the reduction or elimination of oxygen to avoid contact with oils, and preventing exposure to light and pro-oxidative metal ions (copper, iron). Furthermore, substances that can inhibit the oxidation process should be added to increase the shelf life of oils. Different antioxidants are used for this purpose; however, due to the health concerns associated with synthetic antioxidants, efforts are being made to significantly reduce the use of these substances, and replace them with natural or nature-identical antioxidants.

The versatile antioxidant activity of natural antioxidants may contribute to reducing the auto-oxidation of vegetable oils rich in triene-structured polyene fatty acids. When introduced into the diet, these antioxidants can scavenge free radicals, and may exert beneficial effects on the human body. With an aim of identifying novel sources of biologically valuable fats, this study developed a new method for extracting oils from blueberry, cranberry, raspberry, and cuckooflower seeds through sonification. The extracts were subjected to chemical and physicochemical analyses to investigate their composition and content of fatty acids, tocopherols, tocotrienols, phytosterols, and volatile compounds. The proposed method is completely safe, and does not require the use of harmful chemicals.

Berries, especially members of families such as *Rosaceae* (strawberries, raspberries, blackberries) and *Ericaceae* (blueberries, cranberries), are important dietary sources of bioactive compounds [8,9]. They have a delicious taste and pleasant aroma. Due to their antioxidant properties, they are economically valuable, and used as an ingredient in functional foods as recommended by nutritionists and food technologists. In the last decade, there has been intensified research on compounds that can protect the human body from free radicals and other active oxygen species. In particular, lipophilic components of plant oils exhibiting antioxidant activity and free radical-scavenging ability have been widely studied. So far, vegetable oils rich in 18-carbon triene-structured PUFA have been used as pharmaceutical preparations packaged in capsules. However, current oil extraction techniques, such as cold pressing in a nitrogen atmosphere and ultrasonographic extraction, allow for obtaining these oils in an almost unchanged state. Oils thus extracted are richer in biologically active compounds, with high nutritional value and antioxidant activity. 

## 2. Results and Discussion

In this study, the peroxide number, which determines the amount of primary oxidation products, of the analyzed oils ranged from 7.39 to 8.78 meq O_2_/kg. The permissible amount of peroxides in cold-pressed oils is much higher than that in refined oils (<5 meq O_2_/kg) [10]. The peroxide value of the cold-pressed oils analyzed in the study did not exceed 10 meq O_2_/kg, which is the value specified in the standards [10]. This suggests that the chemical composition of lipids (fatty acid composition, content of natural antioxidants and pro-oxidants) and therefore the susceptibility to environmental factors determine the oxidative changes occurring in natural, unrefined oils [11]. All the tested oils were characterized by low acid values (Table 1), which proves the low degree of hydrolysis and the small amount of free fatty acids. The acid values of oils reported in other studies were significantly higher. For instance, a study by Dimić et al. [11] showed that blackberry and raspberry oils had an acid value of 6.85–7.05 and 17.18–17.86 mg KOH/g, respectively. Another study [12] presented acid values of 2.71–3.02, 2.60–3.45, and 2.72–3.73 mg KOH/g for cold-pressed strawberry seed oil, raspberry oil, and blackberry oil, which are similar to the values obtained in this study. The tested oils were products of good quality, not exceeding the prescribed norms, with the exception of acid value of raspberry oil, where the values were slightly above normal.

The chemical characteristics of cold-extracted blueberry, cranberry, raspberry, and cuckooflower native seed oils were evaluated to determine their fatty acid composition, positional distribution of fatty acids, and TAG profile. In addition, the effect of minor components, including tocopherols and pigments, on the oxidative stability of the studied oils was investigated. The results of a previous study by Li et al. [9] indicated that all the tested berry seed oils contained significant levels of palmitic acid (C16:0), stearic acid (C18:0), oleic acid (C18:1), LA (C18:2ω-6), and ALA (C18:3ω-3), along with a favorable ratio of ω-6/ω-3 fatty acids (0.2–1.69). Moreover, palmitic acid, stearic acid, oleic acid, and ALA were predominantly distributed on the terminal positions [9]. In the present study, fatty acid analysis (Table 2) revealed a high content of 18:3 *n-3* acid in the analyzed oils as follows: cress—35.03%, raspberry—31.1%, blueberry—28.99%, and cranberry—34.91%. C 18:1 was detected as the predominant species in the tested berry seed oils: cress—28.24%, raspberry—10.99%, blueberry—21.02%, and cranberry—20.23%. 

Vegetable oils with a low n—6/n—3 acid ratio are especially beneficial to human nutrition. Blueberry, cranberry, raspberry, and cuckooflower oils have an n-3/n-6 ratio of 1–2. These oils can be included in the diets of people who are recalcitrant following heart attacks or cardiac surgeries.

Seed oils from berries such as blueberries, raspberries, and cranberries are a rich source of PUFA. These acids are not synthesized in the human body, due to the lack of enzymes required for forming double bonds in the fatty acid chain at a position beyond C-9. In the human body, n-3 and n-6 fatty acids are part of cell membrane phospholipids, and their proportion in tissues mainly depends on the dietary supply. These are also a source of components that are used in the synthesis of biologically active compounds such as prostaglandins. Essential fatty acids are one of the major building blocks of cells, and are known to exhibit antiarrhythmic [13], anticoagulant [14], antiatherosclerotic [15], and anti-inflammatory [13] effects, and improve vascular endothelial function [15].

In addition to fatty acids, oils from blueberry, cranberry, raspberry, and cuckooflower seeds contain a number of antioxidant, anti-inflammatory, antiatherosclerotic, and anticancerous substances, including tocochromanols, carotenoids, flavonoids, phytosterols, and phenolic acids [2,5,7,16,17,18]. Plant fatty acids, which belong to a group of EUFA, are characterized by high biological activity. Phytosterols are an intriguing group of lipid-like compounds. Among them, β-sitosterol, campesterol, stigmasterol, brassicasterol, Δ5-avenasterol, Δ7-stigmasterol, and Δ7-avenasterol have been reported as important plant oil sterols [19,20]. In most vegetable oils, sterols are the main unsaponifiables. The average content of these compounds in vegetable oils ranges from 400 to 800 mg/100 g, but there can be significant differences in the content in some oils [3,21].

As shown in Table 3, cuckooflower is an excellent source of phytosterols (0.71 mg/g campesterol, 0.27 mg/g campestanol, 0.10 mg/g stigmasterol, 0.13 mg/kg brassicasterol, and 1.58 mg/g β-sitosterol). The combined phytosterol content of cress oil was estimated at 14.41 mg/g, which exceeds the typical range of 1–5 mg/g observed in most vegetable oils [22]. Crude commodity vegetable oils found with high phytosterol content were corn (8–22 mg/g) and rapeseed (5–11 mg/g) oils [22]. On the other hand, raspberry, blueberry, and cranberry extracts contained less amount of phytosterols, which include β-sitosterol (2.01, 1.45, and 1.47 mg/g, respectively) and campesterol (at 0.06–0.10 mg/g). 

Johansson et al. [23] found similar phytosterol content in the seeds of wild fruits from Finnish forests. Sitosterol is an important phytosterol that promotes the slow absorption of cholesterol, and thus allows maintaining low levels of total cholesterol in the peripheral blood. Unlike cholesterol, phytosterols exert some positive effects on the human body. They bind bile acids, and reduce total blood cholesterol without affecting the level of high-density lipoprotein cholesterol [24]. Phytosterols have also been reported to inhibit intestinal cancer. They are characterized by anticancer, antioxidant, and cholesterol-reducing properties [17]. In the present study, the content of tocopherols determined in the tested extracts varied widely. The highest amount of tocopherols was found in raspberry (223.49 mg/100 g), followed by γ-tocopherol (130.38 mg/100 g), and α-tocopherol (75.32 mg/100 g). Cress contained a high amount of tocopherols (112. 44 mg/100 g), with γ-tocopherol and α-tocopherol estimated at 107.94 and 1.09 mg/100 g, respectively. The level of vitamin E in blueberry was 4.84 mg/100 g, whereas in cranberries, it was only 1.65 mg/100 g. 

Compared to the results of a previous study by Omah et al. [25], a higher concentration of tocopherols was noted in the obtained raspberry oil, while the tocopherol profile was similar, with the γ isomer found to be dominant (Table 4). The tocopherol content of cranberry, raspberry, blueberry, and cress seed oils was higher compared to commercial tocopherol-rich oils, such as corn and soybean oil (162 and 180 mg/100 g oil, respectively) [26]. The method of extraction can influence the tocopherol content in oils. The refining process can reduce the content of tocopherols by up to 40%. A study on the oxidative stability of oils [27] indicated that it is difficult to stabilize vegetable oils by adding tocopherols, as native tocopherols in these oils are at the optimal levels needed for stabilization. 

The analysis of volatile odorous compounds revealed that their content was high, and varied depending on the studied plant. In cress, a total of 140 compounds were identified, with dimethyl sulfoxide, 2-butenal, and 2-pentenal found to be dominant. In raspberry, a total of 137 volatile compounds were identified, with hexanal, ethyl acetate, butanoic ethyl ester, 3-hexen-1-ol, and acetate dominant. In blueberry, 173 volatile compounds were identified, of which hexanal, pentanal, cyclopropane 1,1-dimethyl-, benzaldehyde, 5-hepten-2-one, and 6-methyl- were dominant. Cranberry contained 133 volatile compounds, of which 1,2-dimethyl cyclopropane, 1,3,5,7-cyclooctatetraene, 2,4-heptadienal, 2-butenal, 2-heptenal, 2-pentenal, furan, 2-ethyl-, and hexanal were found at high concentrations. 

Table 5, Table 6, Table 7 and Table 8 present the results of the analysis of the most important volatile compounds in the studied oils, whose percentage was above 1%. Among the tested oils, the highest amount of volatile compounds was found in cress oil (30%), cranberry oil (22%), and raspberry and blueberry oil (19%). The main chemical groups represented by the compounds identified were esters, aldehydes, ketones, acids, and terpenes. The high content of esterified volatile aromatic compounds contributes to the fruity characteristic, taste, and aroma of raspberry and cranberry oils. Many of the identified aromatic compounds are characterized by a strong olfactory property, and enhance the characteristic aroma of berries with their spicy, floral, and fruity flavors. This suggests that the aroma of vegetable oils is rendered by the volatile compounds present in the plant materials, such as short-chain fatty acids, heterocyclic compounds, ketones, alcohols, esters, and aldehydes, or results from the processing or storage of the materials [28].

In cold-pressed oils, most chemical changes occur during storage and are related to the oxidation of unsaturated fatty acids [46]. Therefore, it is recommended that cold-pressed oils should be consumed within 6–12 months of production. Oil deterioration is mainly associated with volatile compounds such as aldehydes, ketones, esters, and furan derivatives, which are formed as a result of lipid oxidation processes [47,48], especially the auto-oxidation of unsaturated fatty acids. Several compounds, including hexanal or nonanal, are used as markers to detect the oxidation of lipids [49,50].

The above results highlight that fruit seeds and their oils obtained as agricultural by-products are promising sources of valuable lipid compounds. The seeds are characterized by high oil content, beneficial lipid composition, and high biological activity, as well as potential biotechnological, nutritional, and pharmaceutical applications.

## 3. Materials and Methods

### 3.1. Materials and Extraction of Fat from the Seeds

#### 3.1.1. Sample Preparation

Blueberry, cranberry, raspberry, and cuckooflower seeds were procured from Polfeed in 2020 (Skrzyńsko, Poland), which specializes in the drying and packaging of fruit pomace resulting from the extraction of fruit and vegetable juices. Pomace with a moisture content of approximately 55% was obtained from Hortex (Skrzyńsko, Poland)), and dried on drum driers to reduce the moisture to <10%. The dried pomace was then cut and ground, and the seeds were separated. The production line (Scorpion, Poland), which included a chopper, a separator, and a pneumatic tunnel, was used to separate seeds from other parts. Before extraction, the seeds were ground in a Fritsch PULVERISETTE 25 laboratory mill (Fritsch, Kastl, Germany) with a mesh diameter of 1 × 1 mm.

#### 3.1.2. Extraction Procedures

The extracts from blueberry, cranberry, raspberry, cuckooflower seeds were obtained from the ground material by sonication. The extraction conditions involved the grinding of the seed samples, followed by pre-extraction with 70% ethanol at a temperature of −60 °C for 30 min. After this time as the extraction mixture reached −30 °C the sonification process was started. Since the temperature increased during the sonication process, the vessel with the extraction mixture was placed in ice. A low extraction temperature was used, following previous studies, where it was found that the optimum range for extraction of substances from stone fruit seeds is between −30 °C and −20 °C. The solution was extracted on a Hielscher Ultrasonics UP200ht sonifier (Hielscher, Teltow, Germany) for 15 min under the following conditions: 30 Ws/mL and 90 μm. After the alcoholic extracts were obtained, the solvent was evaporated under nitrogen at 30 °C. The oil layer was collected from the surface, placed in sealed dark glass tubes under nitrogen, and stored in a refrigerator at 4 °C until analysis.

### 3.2. Fatty Acid Composition

The fatty acid composition of the oils was determined by the gas chromatography flame-ionization detection (GC-FID) technique. Fatty acid methyl esters (FAME) were obtained [51] and subjected to GC-FID using a Hewlett-Packard 5890 II (Agilent Technologies, Poway, CA, USA) gas chromatograph equipped with a Supelcowax-10 capillary column (30 m × 0.25 mm × 0.25µm). The initial oven temperature was set at 60 °C, which was then increased to 12 °C/min to 200 °C, and held for 25 min. The injector and detector were maintained at a temperature of 240 °C. The separated FAME were identified by comparing with the retention data of standards. Analyses were performed in three replicates.

### 3.3. Volatile Compounds

The volatile compounds of oils were extracted using a 2-cm CAR/DVB/PDMS fiber (Supelco, Bellefonte, PA) [52]. Samples were in a heater block set on 50 °C for 5 min. Then, the SPME fibre was exposed to a headspace of the sample for 30 min. Next, the SPME fiber was desorbed into the GC injection port at 250 °C. Compounds were identified using an Agilent Technologies 6890N GC×GC-ToF MS system (Agilent Technologies, Palo Alto, CA) coupled to PEGASUS 4 time-of-flight mass spectrometer (LECO, St. Joseph, MI, USA). Separation was performed using two capillary columns: a nonpolar DB-5 column (30 m × 250 mm × 0.5 mm) as the first dimension, and a polar Supelcowax-10 column (0.75 m × 100 mm × 0.1 mm) as the second dimension. The oven initial temperature was 40 °C, and it was held for 3 min. Then, the temperature increased at 4 °C/min to 160 °C. After that, the temperature increased at 10 °C/min to 280 °C, and it was held for 3 min. An electron impact mode was 70 eV, and masses were scanned from 33 to 333 Da. Data were collected and processed using LECO Chroma-TOF v.4.40.

### 3.4. Tocopherols

The qualitative and quantitative determination of tocopherols were carried out using a Waters HPLC system (Waters, Milford, MA, USA), consisting of a pump (Waters 600), a fluorimetric detector (Waters 474), a photodiode array detector (Waters 2998 PDA), an autosampler (Waters 2707), a column oven (Waters Jetstream 2 Plus), and a LiChrosorb Si 60 column (250 × 4.6 mm × 5 µm) from Merck (Darmstadt, Germany). A mixture of n-hexane and 1,4-dioxane (96:4, *v*/*v*) was used as the mobile phase at a flow rate of 1.0 mL/min. The fluorescence of tocochromanols was detected at an excitation wavelength of 295 nm and an emission wavelength of 330 nm [53].

### 3.5. Sterols

The content of sterols was determined by GC following a previously described procedure [54]. Briefly, lipids (0.05 g) were saponified with 1 M KOH in methanol, and the unsaponifiables were extracted using a mixture of hexane and methyl *tert*-butyl ether (1:1, *v*/*v*). The solvent was evaporated under a nitrogen stream, and dry residues were dissolved in anhydrous pyridine, and silylated with BSTFA + 1% TMCS (Supelco, Bellefonte, PA). The sterol derivatives were separated on an Agilent Technologies 6890 Plus GC (Agilent Technologies. Palo Alto. CA, USA) system equipped with a flame-ionization detector and a DB-35MS capillary column (25 m × 0.20 mm, 0.33 μm; Agilent J&W, USA). Samples were injected in splitless mode. The column temperature was initially set at 100 °C and held for 5 min, then increased to 250 °C at 25 °C/min and held for 1 min, and further increased to 290 °C at 3 °C/min and held for 20 min. The detector was set at a temperature of 300 °C. Hydrogen was used as the carrier gas at a flow rate of 1.5 mL/min. 5α-Cholestane was used as an internal standard. Identification was performed by comparing the retention data of compounds with the standards. Samples of each oil were analyzed in triplicate.

### 3.6. Acid Value

The acid value was calculated as the milligrams of potassium hydroxide needed to neutralize the free fatty acids in 1 g of the sample [55].

### 3.7. Peroxide Value

The peroxide value was determined as described by AOCS [56]. 

### 3.8. Statistical Analysis

The analysis of the chemical composition, bioactive compound content, and physicochemical properties of the oils was performed in three replications. The results were expressed as mean and standard deviation.

## 4. Conclusions

To sum up, the oils obtained from blueberry, cranberry, raspberry, and cress seeds by sonification had a favorable fatty acid composition, and significant amounts of tocopherols and phenolic compounds. Due to the high availability of pomace from fruit processing, there is a growing interest in oil sourcing by different companies. The measured parameters (physicochemical properties, fatty acid and volatile compounds composition, and sterols and tocochromanols content) of oils are within a normal range. It shows that sonification is a new efficient technology that can not only be used safely and is environmentally friendly, but is also effective and economical.

## Figures and Tables

**Table 1 molecules-26-07446-t001:** Physicochemical properties of blueberry, cranberry, raspberry, cuckooflower seed oils obtained by sonification.

Characteristics	Blueberry Seed Oil	Cranberry Seed Oil	Raspberry Seed Oil	Cuckooflower Seed Oil
Content of oil (% dry matter)	18.06 ± 0.32	14.52 ± 0.31	16.20 ± 0.29	20.32 ± 0.28
Moisture content of seeds (%)	8.46 ± 0.11	7.76 ± 0.10	7.82 ± 0.11	9.04 ± 0.13
Acid value (mg KOH/g)	2.14 ± 0.00	1.78 ± 0.00	4.14 ± 0.01	1.38 ± 0.00
Peroxide value (meq O_2_/kg)	8.78 ± 0.02	7.39 ± 0.01	8.45 ± 0.03	8.59 ± 0.04

**Table 2 molecules-26-07446-t002:** Fatty acid composition of blueberry, cranberry, raspberry and cuckooflower seed oils obtained by sonification (%).

Fatty Acids	Blueberry Seed Oil	Cranberry Seed Oil	Raspberry Seed Oil	Cuckooflower Seed Oil
C14:0	0	0	0	0.07 ± 0.00
C16:0	4.57 ± 0.10	5.32 ± 0.08	2.31 ± 0.12	7.09 ± 0.10
C16:1	0.06 ± 0.00	0	0	0.09 ± 0.00
C18:0	1.45 ± 0.06	0.87 ± 0.03	0.76 ± 0.03	2.41 ± 0.05
C18:1	21.02 ± 0.48	20.23 ± 0.44	10.99 ± 0.22	28.24 ± 0.36
C18:2n–6	43.33 ± 0.74	37.43 ± 0.62	52.59 ± 0.89	7.05 ± 0.06
C20:0	0.14 ± 0.00	0	0.26 ± 0.01	2.82 ± 0.06
C18:3 n—6	0.08 ± 0.00	0	0	0.12 ± 0.01
C20:1	0.13 ± 0.00	0.15 ± 0.00	0	12.5 ± 0.19
C18:3 n—3	28.99 ± 0.52	34.91 ± 0.59	31.1 ± 0.45	35.03 ± 0.68
C22:1	0	0	0	4.57 ± 0.15
C20:4 n—6	1.08 ± 0.08	1.08 ± 0.08	0	0

**Table 3 molecules-26-07446-t003:** Sterols content of oils from of blueberry, cranberry, raspberry and cuckooflower seed oils obtained by sonification (mg/g).

Phytosterols	Blueberry Seed Oil	Cranberry Seed Oil	Raspberry Seed Oil	Cuckooflower Seed Oil
Cholesterol	0	0	0	0.19 ± 0.01
Brassicasterol	0	0	0	0.12 ± 0.02
Campesterol	0.08 ± 0.01	0.08 ± 0.03	0.11 ± 0.00	0.66 ± 0.06
Campestanol	0	0	0	0.25 ± 0.03
Stigmasterol	0	0.03 ± 0.01	0.06 ± 0.02	0.08 ± 0.03
β-Sitosterol	1.38 ± 0.09	1.51 ± 0.05	1.89 ± 0.18	1.48 ± 0.14
Sitostanol	0.04 ± 0.00	0.04 ± 0.00	0.08 ± 0.03	0.01 ± 0.00
Δ5-Avenasterol	0.03 ± 0.01	0	0.06 ± 0.00	0.60 ± 0.06
Lanosadienol	0.07 ± 0.01	0	0	0
α-amyrin	0	0.03 ± 0.00	0	0
Δ7-Stigmasterol	0.05 ± 0.01	0.39 ± 0.02	0.06 ± 0.01	0
Stigmasta-(8,24)-dien-3β-ol	0.17 ± 0.01	0	0	0
Cycloartenol	0.06 ± 0.01	0	0.10 ± 0.02	0
Δ7-Avenasterol	0	0	0.07 ± 0.00	0
24-methylenecycloartenol	0.08 ± 0.02	0.07 ± 0.00	0.23 ± 0.03	0
Citrostadienol	0	0	0.07 ± 0.01	0
Total content	3460.0	6824.9	4643.1	5384.1

**Table 4 molecules-26-07446-t004:** Tocochromanols content of blueberry, cranberry, raspberry and cuckooflower seed oils obtained by sonification (mg/100 g).

Tocochromanols	Blueberry Seed Oil	Cranberry Seed Oil	Raspberry Seed Oil	Cuckooflower Seed Oil
Tocopherols (mg/100 g):				
α-tocopherol	0.06 ± 0.01	0.09 ± 0.01	75.32 ± 0.42	1.08 ± 0.15
β-tocopherol	nd	nd	nd	nd
γ-tocopherol	0.39 ± 0.01	0.12 ± 0.02	130.38 ± 0.39	107.94 ± 0.41
δ-tocopherol	0.13 ± 0.02	0.06 ± 0.06	18.41 ± 4.6	3.41 ± 0.24
Tocotrienols (mg/100 g): α-tocotrienol	0.07 ± 0.01	0.11 ± 0.01	2.08 ± 0.06	nd
β-tocotrienol	nd	0.14 ± 0.01	1.38 ± 0.13	nd
γ-tocotrienol	2.68 ± 0.05	0.55 ± 0.01	0.33 ± 0.01	nd
δ-tocotrienol	nd	0.60 ± 0.01	nd	nd
Plastochromanol-8	0	0	0	4.00 ± 0.06
Sum of tocols (mg/100 g)	4.85 ± 0.11	1.65 ± 0.01	223.49 ± 0.96	112.44 ± 0.80

nd (not detected)—limit of quantitative for each tocol is 0.5 mg/100 g.

**Table 5 molecules-26-07446-t005:** Percentage composition of volatile compounds from blueberry seed oil obtained by sonification.

Item	RT	Area %	Odor Description	References
3-Methyl-1,4-heptadiene	725	1.09 ± 0.07		
1H-1,2,3-Triazole		1.08 ± 0.06		
2,4-Heptadienal, (E,E)-	999	1.04 ± 0.10	fatty, rancid	[29]
2-Heptenal, (Z)-	960	1.25 ± 0.11	oxidized, tallowy, pungent	[30]
2-Octene	817	1.19 ± 0.10		
2-Pentenal, (E)-	353	1.14 ± 0.04	fruity aroma	[31]
Acetic acid	609	1.53 ± 0.13	sour vinegary	[29]
Acetic acid, anhydride with formic acid	645	1.20 ± 0.09		
Acetic acid, methyl ester	513	2.17 ± 0.15	ethereal, sweet	[29]
Benzaldehyde		1.69 ± 0.14	bitter almonds, penetrating	[29]
2-Nitrobutane		1.20 ± 0.08		
1,1-Dimethylcyclopropane	460	2.46 ± 0.18		
Ethyl Acetate	610	4.26 ± 0.30	sticky, sweet	[30]
Hexanal	801	2.05 ± 0.15	grassy, fatty	[29]
Hexane	600	3.81 ± 0.15	petroleum-like	[32]
Methacrolein	568	3.95 ± 0.28	smell of ozone	[33]
Trimethylene oxide		1.71 ± 0.15	agreeable aromatic	[34]

**Table 6 molecules-26-07446-t006:** Percentage composition of volatile compounds from raspberry seed oil obtained by sonification.

Item	RT	Area %	Odor Description	References
3,5-Heptatriene, (E,E)-		2.19 ± 0.09		
1,6-Heptadien-4-ol	876	0.66 ± 0.05		
3-Methyl-1 butanol	736	2.51 ± 0.22	woody, whiskey, sweet	[29]
3-Methyl-1-butanol, acetate	878	1.13 ± 0.10	banana	[35]
2-Methyl-1-propanol	622	1.94 ± 0.14	wine, penetrating	[29]
2-Hexenal, (E)-	854	1.02 ± 0.07	green leaf, apple-like	[29]
3-Methyl-3-Buten-1-ol	734	1.30 ± 0.09	fresh, fruity, green, slight lavender	[36]
3-Hexen-1-ol, acetate, (Z)-	1009	1.05 ± 0.07		
Acetic acid	609	2.17 ± 0.16	sour vinegary	[29]
Acetic acid, methyl ester	513	1.08 ± 0.06	ethereal, sweet	[29]
Benzene		1.25 ± 0.08	petroleum-like	[37]
Ethanol		1.60 ± 0.07	alcohol	[30]
Ethyl Acetate		8.78 ± 0.52	fruity	[38]
Hexanal	801	2.48 ± 0.21	grassy, fatty	[29]
2,2-Dimethylpropanal	110	3.14 ± 0.29		
Propylene oxide		1.20 ± 0.08	ethereal	[39]
p-Xylene		1.22 ± 0.09	sweet	[40]

**Table 7 molecules-26-07446-t007:** Percentage composition of volatile compounds from cranberry seed oil obtained by sonification Percentage composition of volatile compounds from cranberry seed oil obtained by sonification.

Item	RT	Area %	Odor Description	References
1,2-Dimethyl cyclopropene	539	1.11 ± 0.10		
1,3,5,7-Cyclooctatetraene	850	1.19 ± 0.10		
1-Penten-3-one	684	1.08 ± 0.09	green, pungent	[29]
2,4-Heptadienal, (E,E)-	999	2.27 ± 0.18	fatty, rancid	
2-Butenal, (E)-	649	2.43 ± 0.21	flower	[29]
2-Heptenal, (Z)-	964	1.13 ± 0.08	oxidized, tallowy, pungent	[30]
2-Hexenal	854	1.18 ± 0.07	green leaf, apple-like	[29]
2-Penten-1-ol, (E)-	510	1.87 ± 0.08		
2-Pentenal, (E)-	751	1.78 ± 0.11	fruity aroma	[31]
2-Pentene, (E)-	501	1.65 ± 0.14		
4-Heptenal, (Z)-	888	1.07 ± 0.05		
Acetic acid	609	3.02 ± 0.24	sour, vinegary	[29]
Ethyl Acetate		3.27 ± 0.21	fruity	[38]
2-Ethylfuran	739	1.49 ± 0.06	sweet, etheral	[29]
Furfural	830	2.48 ± 0.22		
Hexanal	801	2.31 ± 0.20	grassy, fatty	[29]
Hexane	600	4.50 ± 0.36	petroleum-like	[32]
Oxirane, methyl-, (S)-		1.44 ± 0.12	sweet, alcoholic	[41]
Pentanal	714	1.27 ± 0.08	woody, bitter, oily	[29]
Trimethylene oxide		1.34 ± 0.10		

**Table 8 molecules-26-07446-t008:** Percentage composition of volatile compounds from cuckooflower seed oil obtained by sonification.

Item	RT	Area %	Odor Description	References
3-Methyl-1 butanol	736	2.09 ± 0.12	disagreeable	
3-Methyl-1 butanol, acetate	867	1.03 ± 0.09	banana	[35]
1-Hexanol	872	1.04 ± 0.08	fruit, soft	[29]
1-Octanol	1078	1.29 ± 0.08	penetrating, aromatic	[42]
1-Penten-3-one	683	1.72 ± 0.12	green, pungent	[29]
2-Methyl-1-propanol	654	2.30 ± 0.13	Wine, penetrating	[29]
2,4-Heptadienal, (E,E)-	999	2.05 ± 0.18	fatty, rancid	[29]
2-Butenal, (E)-	648	4.79 ± 0.33	flower	[29]
2-Ethyl-trans-2-butenal	856	1.69 ± 0.13		
2-Heptanone	889	1.12 ± 0.10	sweet, fruity	[29]
2-Heptenal, (Z)-	964	1.70 ± 0.15	oxidized, tallowy, pungent	[30]
2-Hexenal	855	1.11 ± 0.09	green leaf, apple-like	[29]
2-Pentenal, (E)-	383	1.30 ± 0.80	fruity aroma	[31]
6-Methyl-5-hepten-2-one	988	1.42 ± 0.13	oily, pungent	[29]
Acetic acid, methyl ester	513	1.23 ± 0.01	ethereal, sweet	[29]
Acetonitrile		1.24 ± 0.11	sweet, ethereal	[43]
Carbon dioxide		1.11 ± 0.09	no odor	[44]
Ethyl Acetate	610	4.63 ± 0.32	fruity	[38]
Furan, 2-ethyl-	720	1.18 ± 0.08	smoky, burnt	[45]
Furfural	830	1.14 ± 0.09		
5-Ethyl-2,2,3-trimethylheptan		1.19 ± 0.10		
Hexanal	802	2.67 ± 0.22	grassy, fatty	[27]
Fluorotrinitromethane		1.10 ± 0.08		
3,3-Dimethyloxetane		1.11 ± 0.11		
Pentanal	699	1.41 ± 0.14	woody, bitter, oily	[29]
Trimethylene oxide		1.79 ± 0.12	agreeable aromatic	[34]
p-Xylene		1.09 ± 0.01	sweet	[40]

## Data Availability

Repository of Centre for Open Science.

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
