# Peer review of "Physicochemical Properties, Fatty Acid Composition, Volatile Compounds of Blueberries, Cranberries, Raspberries, and Cuckooflower Seeds Obtained Using Sonication Method"

_molecules, 2021, doi:10.3390/molecules26247446_

Round 1

Reviewer 1 Report

Thank you for submitting the manuscript “Innovative and Pro-ecological Method to Obtain Bioactive Substances From Blueberries, Cranberries, Raspberries and Cuckooflower Seeds to Enrich Functional Foods” to Molecules. Overall, the subject is interesting, but the focus of the work is a little confused. The manuscript needs reworking before it can be considered for the Molecules audience.
1)    Keywords: avoid words that have already been included in the manuscript title 
2)    My biggest concern regarding the manuscript is related to the fact that the authors report that it is a new and pro-ecological method. As this information was given right in the title, it would be expected that a standard method was used as a comparison and this ended up impoverishing the results obtained. 
3)    In the abstract, there is a very strong tendency to extrapolate interpretations in relation to the results obtained (lines 29-31). I believe that the best thing is for the authors to keep their interpretations and conclusions related to the results obtained in this work. The same happened in the title where the expression “to enrich functional foods” makes the reader understand that the manuscript will address an application for the bioactive compounds obtained from the seed. 
4)    The results related to volatile compounds are beyond the scope of this work since, in general, these compounds compose aroma and are not necessarily related to beneficial effects on human health. If there is any such claim, this point of view should be defended so that the volatile compound results fit better in the manuscript. 
5)    Minor errors:
Line#108-112: this manuscript did not address the use of berries as a fruit only from their seeds. I do not believe this statement is convenient for opening the material and methods item.
Line#108-121: if the authors believe this paragraph is important it should appear in the introduction. For the item results, the results are expected to be addressed and discussed. I don't see how this paragraph fits into this item.
Line#123: “mil 02/kg” Is this unit of measurement correct? Also separate symbols from numbers throughout the text.
Line#132, #140, #210, #358, etc: correct the citation form

Author Response

Comments and Suggestions for Authors

Thank you for submitting the manuscript “Innovative and Pro-ecological Method to Obtain Bioactive Substances From Blueberries, Cranberries, Raspberries and Cuckooflower Seeds to Enrich Functional Foods” to Molecules. Overall, the subject is interesting, but the focus of the work is a little confused. The manuscript needs reworking before it can be considered for the Molecules audience.
1)    Keywords: avoid words that have already been included in the manuscript title 

The keywords were corrected.

2)    My biggest concern regarding the manuscript is related to the fact that the authors report that it is a new and pro-ecological method. As this information was given right in the title, it would be expected that a standard method was used as a comparison and this ended up impoverishing the results obtained. 

The tittle was changed so that it suits better to the content of the paper.

3)    In the abstract, there is a very strong tendency to extrapolate interpretations in relation to the results obtained (lines 29-31). I believe that the best thing is for the authors to keep their interpretations and conclusions related to the results obtained in this work. The same happened in the title where the expression “to enrich functional foods” makes the reader understand that the manuscript will address an application for the bioactive compounds obtained from the seed. 

The tittle and abstract were changed according to the recommendations.

4)    The results related to volatile compounds are beyond the scope of this work since, in general, these compounds compose aroma and are not necessarily related to beneficial effects on human health. If there is any such claim, this point of view should be defended so that the volatile compound results fit better in the manuscript. 

The title and abstract were changed suggesting the content of the paper in a more correct way and paying attention to the fact that the assessment of volatile compounds was an important element of the study.

5)    Minor errors:
Line#108-112: this manuscript did not address the use of berries as a fruit only from their seeds. I do not believe this statement is convenient for opening the material and methods item.

The statement was removed from material and methods item.

Line#108-121: if the authors believe this paragraph is important it should appear in the introduction. For the item results, the results are expected to be addressed and discussed. I don't see how this paragraph fits into this item.

This paragraph was moved to the “Introduction” section.

Line#123: “mil 02/kg” Is this unit of measurement correct? Also separate symbols from numbers throughout the text.

The unit was changed, and the separation were added.

Line#132, #140, #210, #358, etc: correct the citation form

The citations were corrected.

Reviewer 2 Report

The manuscript is an original and interesting study.  However, some corrections must be made to improve the quality of the article and, consequently, allow its publication. Formal English must be revised and the structure of the manuscript and the discussion need to be improved.

# Abstract:

  • The keywords must be revised. "Raspberry and cuckoo flower native seed oils" is a very extensive keyword.

# Introduction:

  • The first paragraph of the introduction is very extensive. I strongly advise authors to summarize this paragraph in their main topics.

# Results and discussion:

  • Line 108 to line 121 does not fit into results and discussion. This paragraph is about the introduction.

  • Line 123: the unit must be revised.

  • Second paragraph: Are the results observed favorable ​​or not? Make it clearer in the text.

  • Third paragraph: What is the relationship between the composition of oils and the parameters previously evaluated?

  • Fourth and fifth paragraphs: Again the paragraph does not fit into results and discussion but rather into introduction.

  • Line 248 to 251: This information was already mentioned in the second paragraph of the discussion. It is recommended that the authors suppress this paragraph or complete paragraph 2 by summarizing the information.

  • Line 257: We strongly recommend that authors reorganize the tables, placing them within the text, mentioning them and, soon after, carrying out the discussion. Table 1 provides a comparison between the different raw materials in relation to the evaluated parameters. We suggest that the authors perform a statistical test to show whether the differences between the results are significant or not. For example, Tukey test.

# Material and methods: I recommend that the authors divided section 3.1 into two sections. The first about sample preparation and the second about extraction procedures.

  • Line 313: Insert the date of sample collection.

  • Line 319: Add in parentheses: model and brand, city and country. Check the entire manuscript.

  • Line 322: How was the pre-extraction performed and why at such a low temperature (-60 °C)? Besides that, separate unit of the value.

  • Make it clear what the parameters mentioned in line 324 are and how they were determined.

  • Line 375: I suggest that authors briefly describe the method adopted.

  • How were the aroma sensory analyzes mentioned in the abstract performed?

  • At various moments in the text, the authors mention the antioxidant potential of raw materials. Why did not they perform antioxidant activity analysis (DPPH, FRAP, or ABTS) to confirm these claims?

# The conclusion can be improved, giving more value to the results obtained. Besides that, make it clearer why these compounds are of interest to the food and cosmetics industries (mentioned in the conclusion).

# References: There is a format error in references 17 and 18. References 37 to 59 must be observed and placed in standard format.

Author Response

Comments and Suggestions for Authors

The manuscript is an original and interesting study.  However, some corrections must be made to improve the quality of the article and, consequently, allow its publication. Formal English must be revised and the structure of the manuscript and the discussion need to be improved.

The language of the manuscript was corrected.

# Abstract:

  • The keywords must be revised. "Raspberry and cuckoo flower native seed oils" is a very extensive keyword.

The keywords were corrected.

# Introduction:

  • The first paragraph of the introduction is very extensive. I strongly advise authors to summarize this paragraph in their main topics.

The paragraph was shortened.

# Results and discussion:

  • Line 108 to line 121 does not fit into results and discussion. This paragraph is about the introduction.

This paragraph was moved to the “Introduction” section.

  • Line 123: the unit must be revised.

The unit was changed.

  • Second paragraph: Are the results observed favorable ​​or not? Make it clearer in the tex.

The proper sentence was added to the paragraph.

  • Third paragraph: What is the relationship between the composition of oils and the parameters previously evaluated?

Whether this question could be clarified because it is not entirely clear to us?

  • Fourth and fifth paragraphs: Again the paragraph does not fit into results and discussion but rather into introduction.

These paragraphs have been significantly shortened and merged into one so that their content does not differ from the results and discoussion chapter but at the same time it emphasizes the importance of fatty acids in the human diet.

  • Line 248 to 251: This information was already mentioned in the second paragraph of the discussion. It is recommended that the authors suppress this paragraph or complete paragraph 2 by summarizing the information.

The information was removed.

  • Line 257: We strongly recommend that authors reorganize the tables, placing them within the text, mentioning them and, soon after, carrying out the discussion. Table 1 provides a comparison between the different raw materials in relation to the evaluated parameters. We suggest that the authors perform a statistical test to show whether the differences between the results are significant or not. For example, Tukey test.

According to the suggestion the tables were placed within the text. Statistic is not necessary because the work was not intended to show differences between specific oils, but only to show the results of obtaining material by sonication. Same as it is in the tables below.

# Material and methods: I recommend that the authors divided section 3.1 into two sections. The first about sample preparation and the second about extraction procedures.

The section was divided.

  • Line 313: Insert the date of sample collection.

This information was completed in the manuscript

  • Line 319: Add in parentheses: model and brand, city and country. Check the entire manuscript.

This information was completed in the manuscript.

  • Line 322: How was the pre-extraction performed and why at such a low temperature (-60 °C)? Besides that, separate unit of the value.

Initial extraction was performed by flooding the samples with 70% ethyl alcohol at -60 ° C and, after mixing, left at room temperature for about 30 minutes. After the extraction mixture had reached -30 °C, the sonication process was started. A low temperature was used, guided by previous studies, where it was found that the optimal range of substance extraction from stone fruit seeds is between -30 ° C and -20 ° C.

  • Make it clear what the parameters mentioned in line 324 are and how they were determined.

Whether this question could be clarified because it is not entirely clear to us, especially that line 324 is the middle of the table?

  • Line 375: I suggest that authors briefly describe the method adopted.

This information was completed in the manuscript.

  • How were the aroma sensory analyzes mentioned in the abstract performed?

The abstract was corrected, sensory analyzes were not performed.

At various moments in the text, the authors mention the antioxidant potential of raw materials. Why did not they perform antioxidant activity analysis (DPPH, FRAP, or ABTS) to confirm these claims?

In the research, we focused primarily on the composition of cranberry, blueberry, raspberry, cuckooflower seed oil and basic indicators proving the quality of the oil, including acid value and peroxide value. Surely the designation DPPH, FRAP, or ABTS would enrich the work.

# The conclusion can be improved, giving more value to the results obtained. Besides that, make it clearer why these compounds are of interest to the food and cosmetics industries (mentioned in the conclusion).

The conclusion was improved.

# References: There is a format error in references 17 and 18. References 37 to 59 must be observed and placed in standard format.

The refferences 17 and 18 were corrected. References 37-59 were changed according to the PubChem Citation Guidelines.

Round 2

Reviewer 1 Report

Thank you for submitting the manuscript “The Results Of The Bioactive Substances And Volitale Compounds Content In Blueberries, Cranberries, Raspberries And Cuckooflower Seeds Obtained By New And Safe Sonication Method” to Molecules. Thank you for incorporating the suggested modifications, this made the quality of the manuscript much improved. However, my suggestion for the title of the work is something more objective like “Physicochemical properties, Fatty acid composition, Volitale Compounds of Blueberries, Cranberries, Raspberries And Cuckooflower Seeds Obtained using Sonication Method”

Author Response

The title was changed according to the reviewer suggestion.

Reviewer 2 Report

Innovative and Pro-ecological Method to Obtain Bioactive Substances From Blueberries, Cranberries, Raspberries and Cuckooflower Seeds to Enrich Functional Foods

# Third paragraph: What is the relationship between the composition of oils and the parameters previously evaluated? Whether this question could be clarified because it is not entirely clear to us?

The question is whether the composition of oils affects the responses, peroxide value and acid value and if so how does this occur.

# Whether this question could be clarified because it is not entirely clear to us, especially that line 324 is the middle of the table?

This question refers to the end of section 3.1.2. specifically the part of sonication extraction.

Author Response

# Third paragraph: What is the relationship between the composition of oils and the parameters previously evaluated? Whether this question could be clarified because it is not entirely clear to us?
The question is whether the composition of oils affects the responses, peroxide value and acid value and if so how does this occur.

Answer:
Classic measures of changes in oils are the peroxide number and the acid number. Wp.gp. Peroxide is a peroxide measurement and is treated as an indicator of the degree (rancidness), expressing the number of milliliters of thiose work to be measured. Maximum LOO values ​​in selected fatty fats with fat peroxide fat, rapeseed, palm, coconut 5.01; olive oil 20.02; lard 03 [ meq O2 / kg]. The acid number determines the amount of free fatty acids. It is expressed as the amount of carbon hydroxide needed to neutralize the fatty acids in 1 gram of the test compound. Sour is a measure of the free quality of fatty acids, meaning the price of hydrolysis, the number of free space. Maximum LK values ​​in selected fats is: soybean oil, rapeseed oil, coconut 0.31 olive oil 6.6 (raffin 0.6) lard 1.13).

# Whether this question could be clarified because it is not entirely clear to us, especially that line 324 is the middle of the table?
This question refers to the end of section 3.1.2. specifically the part of sonication extraction.
Answer:
The section was changed according to the suggestion.